# CEEMDAN-ICA-Based Radar Monitoring of Adjacent Multi-Target Vital Signs

**Xichao Dong** [1,2,3,4], **Yun Feng** [1,2,3], **Chang Cui** [1,2,3,*] and **Jun Lu** [1,2]

1   Beijing Institute of Technology Chongqing Innovation Center, Chongqing 401120, China;
    xcdong@bit.edu.cn (X.D.); 3220200535@bit.edu.cn (Y.F.); lujun@bitcq-radarlab.com (J.L.)
2   Chongqing Key Laboratory of Novel Civilian Radar, Chongqing 401120, China
3   School of Information and Electronics, Beijing Institute of Technology, Beijing 100081, China
4   Key Laboratory of Electronic and Information Technology in Satellite Navigation (Beijing Institute of
    Technology), Ministry of Education, Beijing 100081, China
*   Correspondence: changcui@bit.edu.cn

**Abstract:** In recent years, radar, especially frequency-modulated continuous wave (FMCW) radar, has been extensively used in non-contact vital signs (NCVS) research. However, current research does not work when multiple human targets are close to each other, especially when adjacent human targets lie in the same resolution cell. In this paper, a novel method based on complete ensemble empirical mode decomposition with adaptive noise (CEEMDAN)–independent component analysis (ICA) was proposed to obtain the vital-sign information (including respiratory rate and heart rate) of adjacent human targets by using a single FMCW radar. Firstly, the data observed at a single angle were decomposed by the CEEMDAN separation algorithm to construct virtual multi-angle observations. It can effectively transform the undetermined blind source separation (UBSS) problem into an overdetermined blind source separation (BSS) problem. Thus, a BSS algorithm based on FastICA can be used to reconstruct each person's vital-sign signal and then calculate their respiratory rate/heart rate. To validate the effectiveness of the proposed method, experiments based on the measured data were conducted and the results show that the proposed method can obtain multi-target vital-sign information even when they are in the same resolution cell.

**Keywords:** vital signs; FMCW radar; blind source separation

## 1. Introduction

An individual's vital signs, especially respiration rate (RR) and heart rate (HR), are important indicators of the physical health of high-risk groups, such as those with cardiovascular diseases, elders at home, infants, etc. In addition, such micro-movement information can also be used for personal recognition due to its uniqueness [1]. Many types of equipment have been developed to monitor vital signs. One of the most popular observation systems is the frequency-modulated continuous wave (FMCW) radar, which overcomes privacy issues and can meet the needs of long-term real-time observation.

Numerous studies have been conducted to improve the performance of vital signs measurement by FMCW radar. Anitori et al. found that the respiratory and heartbeat information can be calculated from the FMCW signal's phase [2]. On this basis, a hybrid radar system that combines the linear-frequency-modulated continuous-wave (LFMCW) mode and the interference measurement mode was developed to continuously track individuals' vital signs within complex indoor environments [3]. Adib et al. proposed a vital-radio technology to increase the vital sign detection range [4]. In addition, to separating different vital signs with high accuracy, a dual-parameter least mean square (LMS) filter [5], a compressive sensing based on the orthogonal matching pursuit (CS-OMP) method, and a rigrsure adaptive soft threshold noise reduction based on discrete wavelet transform

(RA-DWT) [6] were proposed. However, these methods are only applicable to a single human target and would fail when there are multiple targets.

Since actual application scenarios often involve multiple targets, researchers have improved the methods for separating different targets before vital sign measurement. There are two main methods for multi-target vital-sign monitoring. One is based on target distance information and the other is based on target azimuth information. Lee et al. proposed a target separation method by improving distance resolution [7]. However, they were still unable to separate the vital sign signals of multiple targets located in the same range cell.

In order to separate targets in the same range cell, it is necessary to estimate the angle-of-arrival (AOA) of different targets. This technique can be further divided into hardware-based methods [7–13] and software-based methods [14–17]. Hardware-based methods obtain non-contact vital signs information by designing the radar hardware structure. For example, a mechanical scanning (MS)-based solution achieves beam scanning by rotating the radar antenna. In [9], Islam separated the respiration signals by rotating the radar module and used independent component analysis with joint approximate diagonalization of the eigenmatrices algorithm (ICA-JADE) to separate the respiration signals of two targets located within the beam width. Nosrati et al. developed a 2.4 GHz dual-beam phased array continuous wave multi-input multi-output (MIMO) radar, which use phase shifters for controlling the phase of each receiving channel [13]. In [12], Mingle et al. designed a bidirectional leaky wave antenna (LWA) with beam scanning.

The above methods require prior knowledge of the angle of the measured target. Therefore, software-based methods, such as digital beamforming (DBF), are proposed. The DBF technique multiplies each channel with a weight vector during signal processing to boost the signal strength in a certain direction. Ahmad et al. proposed an FMCW radar featuring three transmitting antennas and four receiving antennas for detecting the respiration signals of multiple people [14]. Xiong et al. proposed a method based on CW radar for the detection of the respiration signals of multiple targets located in the same distance bin [17]. This method estimates the direction-of-arrival (DOA) to obtain the number and angle of each target and then uses adaptive DBF (ADBF) technology to extract respiration signals for specific targets. Kakouche et al. proposed the incoherent signal subspace method (ISSM) and linearly constrained minimum variance (LCMV), which realize a respiratory rate error of about 2% [16]. Additionally, Dang et al. sued an IR-UWB radar and proposed an optimization algorithm, CIR-SS, based on the channel impulse response (CIR) smoothing spline method, achieving through-wall multi-person NCVS detection with an average accuracy of 92.20% [15].

However, these methods can only separate multi-target vital signs that are far away from each other. That is, existing methods cannot be used to detect the vital signs of a couple or a new mother sleeping with her infant in the same bed. The separation of multi-target vital signs with close distances and inseparable azimuth angles still requires further research. When multiple targets cannot be distinguished in the space domain, signal separation can be considered from the signal domain, and the method of blind source separation can be introduced. Qiao et al. achieved the separation of micro-Doppler signals by using short-time fractional Fourier transform (STFrFT) and morphological component analysis (MCA) [18]. Yue et al. designed the DeepBreath system, a radio frequency (RF)-based respiratory monitoring system that can separate respiratory signals even when multiple targets are adjacent [19]. Nevertheless, this system failed to separate multi-target heartbeat signals and relied on the size of the antenna array.

To solve the above problems, this paper proposed a multi-target vital sign monitoring method based on CEEMDAN-ICA, which can separate targets in the same resolution cell with a single FMCW radar. The method models the multi-target vital sign monitoring problem as a blind source separation (BSS) problem, and fully utilizes the different frequency characteristics of different people to convert the underdetermined blind source separation (UBSS) problem into an overdetermined one, enabling the separation of hu-

man bodies with only a single radar. Firstly, this method adopts the complete ensemble empirical mode decomposition algorithm (CEEMDAN) to pre-separate the echo signal. The decomposition results can be regarded as virtual multi-angle observations. Then, the FastICA algorithm was used to separate each person's vital sign signal and calculate their respiratory rate/heart rate based on the spectrum's peak value. Finally, the experimental results show that the proposed method can obtain the multi-target vital sign information even when targets are in the same resolution cell and have high separation accuracy. This paper's contributions are as follows.

1. This paper proposed a multi-target vital-sign signal processing framework that can accurately separate each person's signal even when the people are in the same resolution cell. Compared with traditional methods with a single radar that can only extract single-target information, this method can extract both respiratory rate and heart rate for each person simultaneously.

2. This paper proposed a multi-target vital-sign separation method based on CEEMDAN-ICA. The method innovatively uses the CEEMDAN algorithm to reconstruct multiple angle observations, allowing accurate signal separation with only a single radar system.

The structure of this paper is as follows. Section 2 introduces the multi-target vital signs echo signal model. Section 3 presents the proposed method based on CEEMDAN-ICA for multi-target vital signs detection. Section 4 provides the experimental data processing results and compares different algorithms to demonstrate the superiority of the proposed algorithm. The conclusion is in Section 5.

## 2. Modeling Mixtures of Multi-Target Vital Signs Echo Signal

Consider that there are P stationary human targets at distances $R_{0,1}, R_{0,2}, \ldots, R_{0,P}$. For the $p$-th human target, the FMCW radar echo intermediate frequency (IF) signal can be expressed as:

$$s_p(t) = A_{IF,p} \exp[j(2\pi \frac{2KR_p(t)}{c}t + \frac{4\pi R_p(t)}{\lambda_c})], \tag{1}$$

$$R_p(t) = R_{0,p} + x_p(t), \tag{2}$$

where $x_p(t)$ is the chest displacement information of the $p$-th human target, which contains the information of human breathing and heartbeat movement. Generally, it can be approximated as a sine function:

$$x_p(t) = A_{R,p} \sin(2\pi f_{R,p} t) + A_{H,p} \sin(2\pi f_{H,p} t), \tag{3}$$

where $A_{R,p}$, $A_{H,p}$ are the respiration and heartbeat movement amplitudes of the $p$-th human target and $f_{R,p}$, $f_{H,p}$ are the respiration rate and heart rate of the $p$-th human target, respectively.

The frequency response of the IF signal is:

$$h(R(t), f) = A_{IF,p} T_c \exp(-j2\pi \frac{f_c}{K_c} f) sinc(\frac{R_p(t)c}{2B} + T_c f) \exp(2\pi j(\frac{f_c}{B} + \frac{1}{2})(\frac{R_p(t)c}{2B} + T_c f)); \tag{4}$$

therefore, the frequency domain of the IF signals of P targets can be expressed as:

$$H(R(t), f) = \sum_{i=1}^{P} h(R_i(t), f). \tag{5}$$

From (4) and (5), it can be seen that, when there are multiple targets in the scene, the function $H(R(t), f)$ cannot be expressed as a linear sum of P independent components. Expanding $h(R_i(t), f)$ to $h(R_{0,i} + x_i(t), f)$. Using Taylor series to first-order terms, it can be approximated as

$$h(R_i(t), f) = h(R_{0,i}, f) + x_i(t)h'(R_{0,i}, f), \tag{6}$$

where $h'(R_{0,i}, f)$ is the derivative with respect to range domain. Therefore, the total frequency response of P independent sources (human echo signal) is:

$$H(R(t), f) = \sum_{i=1}^{P} h(R_{0,i}, f) + \sum_{i=1}^{P} h'(R_{0,i}, f) x_i(t). \tag{7}$$

The first term of (7) is just the average frequency response of all sweep time units. So, we can subtract the first term from the model without affecting the vital signs signal. The second term of (7) can be regarded as the model of the BSS method, since $x_i(t)$ is not related to the frequency $f$ and $h'(R_{0,i}, f)$ corresponds to the mixing coefficient for each $f$.

## 3. Multi-Target Vital Signs Detection Method Based on CEEMDAN-ICA

The general ICA method cannot solve the UBSS problem (i.e., the number of receiving channels is less than that of source signals) but can only solve the overdetermined or positively determined situations. In this section, the CEEMDAN method was used to pre-separate the signal. After that, the intrinsic mode functions (IMF) components with a frequency in the range of 0.15–2.0 Hz (i.e., the effective vital sign frequency) were screened to establish a virtual multi-channel, so the UBSS problem can be transformed into an overdetermined/positive one. Then, the FastICA algorithm was used to separate the breathing and heartbeat signals of multi-human targets. The flowchart is shown in Figure 1.

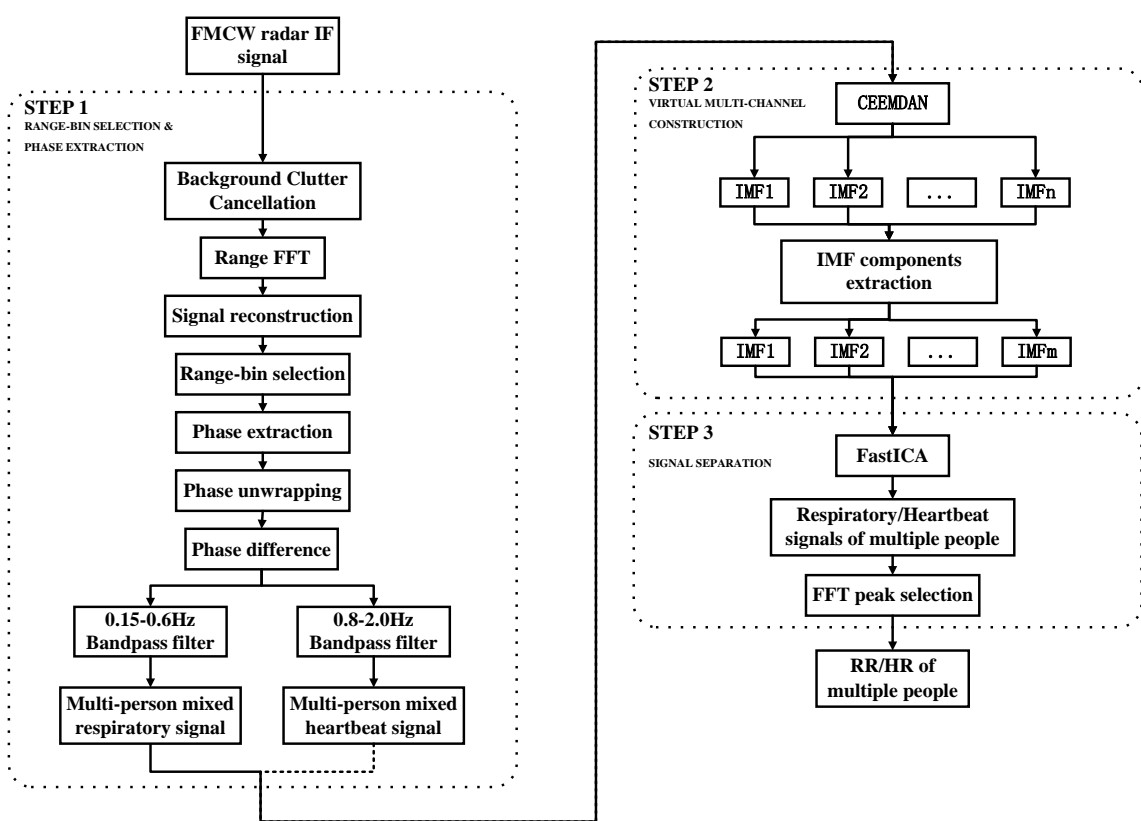

**Figure 1.** Radar multi-adjacent-target vital signs signal separation method based on CEEMDAN-ICA.

### 3.1. Background Clutter Cancellation

3.1.1. Background Clutter Cancellation

For the convenience of subsequent signal processing, the radar echo data need to be reconstructed first. The FMCW millimeter-wave radar echo data were reshaped into a matrix $D(n, m)$ according to the received order of chirps, where $n$ is the fast-time point (i.e., the number of analog-to-digital converter (ADC) sampling points in each chirp), which

correspond to the chirp cycle *Tc* and *m* is the slow-time point (i.e., the number of frames), which correspond to the frame cycle *Tm*.

In actual scenarios, there may exist some random environmental noise and some stationary irrelevant targets, such as walls, tables, and chairs. To eliminate the influence of background/stationary clutter, this paper adopted clutter cancellation: the unmanned scene echo $D_0(n, m)$ was subtracted from the processed echo matrix $D(n, m)$:

$$D(n, m) = D(n, m) - D_0(n, m). \tag{8}$$

This can greatly weaken the influence of background stationary targets.

### 3.1.2. Range-Bin Selection

The range profile information $H(k, m)$ of the target can be obtained by performing K-point fast Fourier transform (FFT) on the discrete radar data in the range direction, where *k* is the FFT point and the position of the peak of the spectrum reflects the distance of the target. According to (7), the average frequency response is obtained for the multi-frame range image information and the reconstructed signal that can be applied to ICA can be calculated as:

$$H_{ICA}(k, m) = H(k, m) - \frac{1}{M}\sum_{m=1}^{M} H(k, m). \tag{9}$$

To effectively extract the phase containing human thoracic displacement information, it is of great necessity to select the effective range cell where the human body is located. As in (10), the energy of data at the same range-bin in each frame (also each chirp) are superimposes and those which have the highest energy are selected. By extracting all the $k_0$ range-bin of each frame, a slow-time signal containing human respiration and heartbeat information can be formed as (11).

$$k_0 = \max(\sum_{i=1}^{M} |H(k, i)|^2) \tag{10}$$

$$M(m) = H_{ICA}(k_0, m). \tag{11}$$

### 3.1.3. Phase Extraction

We can find from (1) that the phase of the intermediate frequency (IF) signal in the radar backscatter contains information on the displacement of the human chest moving (physiological signal). Therefore, the slow-time signal's phase can be written as:

$$\angle M(m) = 4\pi \frac{R_0 + x(mT_m)}{\lambda_c} = \frac{4\pi}{\lambda_c} x(mT_m) + \phi_0. \tag{12}$$

The phase angle radians of each slow-time data can be calculated by:

$$\theta(m) = \arctan(\frac{Im(M(m))}{Re(M(m))}), \tag{13}$$

where $Re(M(m))$ and $Im(M(m))$ are the real and imaginary parts of $M(m)$, respectively.

However, phase wrapping may occur during calculation, so it is necessary to correct, also called phase unwrapping. Specifically, when the absolute value of the phase difference between the current and the previous frames is greater than $\pi$, the phase difference was subtracted or added to obtain a new difference whose absolute value was less than $\pi$. The previous phase was then added to the new difference value to obtain the phase with unwrapping:

$$\theta'(m+1) = \begin{cases} \theta(m+1), & if -\pi \leq \theta(m+1) - \theta(m) \leq \pi \\ \theta(m+1) + 2\pi, & if \ \theta(m+1) - \theta(m) < -\pi \\ \theta(m+1) - 2\pi, & if \ \theta(m+1) - \theta(m) > \pi \end{cases}. \tag{14}$$

As can be seen from (12), the directly calculated phase signal still has a direct current (DC) bias due to the human distance from the radar. To eliminate the DC bias, a first-order difference method can be used, also known as phase difference, that is, subtracting the previous phase from the current phase:

$$x(m) = \theta'(m+1) - \theta'(m), \tag{15}$$

where $x(m)$ is the phase signal containing the human chest movement (respiratory and heartbeat) information. In addition, phase difference can also enhance the amplitude of the heartbeat signal.

The ratio of the amplitude of the heart rate signal to the respiratory signal amplitude in (3) is:

$$\frac{A_{H,p}}{A_{R,p}}. \tag{16}$$

Taking the first-order difference of the physiological signal model, we can get:

$$\Delta x_p(mT_m) = 2\pi f_{R,p} A_{R,p} \cos(2\pi f_{R,p} mT_m) + 2\pi f_{H,p} A_{H,p} \cos(2\pi f_{H,p} mT_m). \tag{17}$$

At this time, the ratio of the heart rate signal amplitude to the respiratory signal amplitude becomes:

$$\frac{2\pi f_{H,p} A_{H,p}}{2\pi f_{R,p} A_{R,p}} = \frac{f_{H,p}}{f_{R,p}} \cdot \frac{A_{H,p}}{A_{R,p}}. \tag{18}$$

Comparing (18) with (16), it can be seen that the ratio has more coefficients. Since the heart rate frequency is generally in the range of 0.8–2.0 Hz and the respiratory rate is in the range of 0.15–0.5 Hz, this coefficient is greater than 1.6, indicating that the heart rate signal is enhanced compared to the respiratory signal

### 3.2. Virtual Multi-Channel Construction

3.2.1. Respiratory/Heart Rate Signal Separation

The respiratory rate of adults is generally within the range of 0.1–0.5 Hz and the heart rate is within the range of 0.8–2.0 Hz. Therefore, separating the respiratory and heart rate signals only requires designing corresponding bandpass filters. General digital filters are mainly divided into finite impulse response (FIR) filters and infinite impulse response (IIR) filters. Considering the low complexity of respiratory/heartbeat signals and focus on amplitude-frequency characteristics, this article selects IIR filters, which have a faster calculation speed.

Commonly used IIR filter design methods include Butterworth, Chebyshev type I, Chebyshev type II, and elliptic. From these four design methods, four different IIR bandpass filters were designed for respiratory and heartbeat signals. Compared to high-order filters, low-order filters can better compromise the performance and robustness of the system and are easy to implement. To reduce calculation complexity, a 6th-order IIR filter with a passband of 0.15–0.6 Hz was designed for respiratory signals and an 8th-order IIR filter with a passband of 0.8–2.0 Hz was designed for heartbeat signals. The magnitude response of the designed filters is shown in Figure 2.

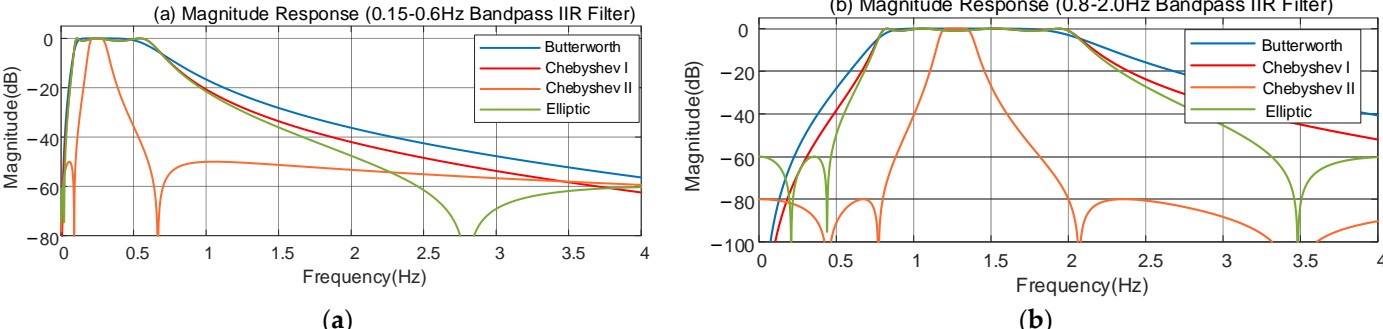

**Figure 2.** The magnitude response of the designed IIR bandpass filters: (**a**) Magnitude response of the filter for respiratory signal (0.15–0.6 Hz); (**b**) Magnitude response of the filter for heartbeat signal (0.8–2.0 Hz.).

Based on the comparison of the magnitude response chart, it can be found that the Chebyshev type II passband is more stable but too narrow, while the Butterworth type is stable but with a wider transition band. Although Chebyshev type I and elliptic passbands have fluctuations, they are wider, and the roll-off speed is faster. The elliptic type has a faster roll-off speed than the Chebyshev type I, so the elliptic type IIR filter is selected.

### 3.2.2. Virtual Multi-Channel Construction Based on CEEMDAN

The complete EEMD with adaptive noise (CEEMDAN) algorithm is a new signal decomposition algorithm that solves the mode mixing phenomenon of the empirical mode decomposition (EEMD) [20]. The principle of the CEEMDAN method is to add white noise (or IMF components with white noise) to the residual after obtaining the first-order IMF component, and to iterate by taking the mean of the IMF component at this time. In this paper, CEEMDAN was used to separate the mixed signal to obtain multiple IMF components and then screen them to construct a virtual multi-channel. This method sequentially processed the filtered signal obtained in the previous section, and the processing steps were as follows.

Step 1: Add Gaussian white noise with a mean of 0 and an amplitude of $\varepsilon$ to the signal to be decomposed, such as (19), to construct the I signals to be decomposed.

$$x_i(m) = x(m) + \varepsilon\delta_i(m), i = 1, 2, \ldots, I.; \tag{19}$$

Step 2: Perform empirical mode decomposition (EMD) decomposition on $x_i(m)$ in Step 1 to obtain the first IMF component and take the average as the first IMF component of CEEMDAN, such as (20), where $IMF_1(m)$ is the first IMF component and $r_1(m)$ is the residual signal after the first decomposition.

$$IMF_1(m) = \frac{1}{I}\sum_{i=1}^{I} IMF_1^i(m)$$
$$r_1(m) = x(t) - IMF_1(m) \tag{20}$$

Step 3: Add specific noise to the residual signal decomposed after the $k$th iteration and continue to EMD decomposition, such as (21), where $IMF_k(m)$ is the $k$th IMF component obtained by CEEMDAN decomposition; $E_{k-1}(\cdot)$ is the $k-1$th IMF component obtained by EMD decomposition; $\varepsilon_{k-1}$ is the coefficient of adding noise to the residual signal during the $k-1$th iteration of CEEMDAN; and $r_k(m)$ is the residual signal after the $k$th iteration.

$$IMF_k(m) = \frac{1}{I}\sum_{i=1}^{I} E_1(r_{k-1}(t) + \varepsilon_{k-1}E_{k-1}(\delta_i(m)))$$
$$r_k(m) = r_{k-1}(m) - IMF_k(m) \tag{21}$$

Step 4: If the residual signal $r_k(m)$ after the Kth decomposition is a monotonic signal, stop the iteration. If it is not satisfied, return to Step 3;

Step 5: The obtained IMFs are used to perform FFT and obtain the peak value to obtain the IMF frequency. Selecting IMFs with a frequency in the 0.15–0.6 Hz range for multi-human respiratory signals separation, and IMFs with a frequency in the 0.8–2.0 Hz for multi-human heartbeat signals separation. This Step can reduce the dimensions of IMFs and can also remove low/high frequency noise.

After the above steps, virtual multi-channels containing breathing/heartbeat information of multiple people can be constructed.

### 3.3. Vital Signs Signal Separation Based on FastICA

To obtain breathing/heartbeat information for each individual, an independent component analysis (ICA) can be performed on the mixed signal. Before applying the ICA algorithm to the data, a series of preprocessing steps can be performed to simplify the calculations, including centering and whitening.

Centering is performed to simplify the ICA algorithm; the data can be zero-centered, i.e., the mixed data are subtracted from its mean vector.

Whitening is a linear transformation of the observed mixed variables $\mathbf{x}$, resulting in a whitened vector $\widetilde{\mathbf{x}}$ with unit variance and uncorrelated components. The covariance matrix is a unit matrix, i.e., $E\left\{\widetilde{\mathbf{x}}\widetilde{\mathbf{x}}^T\right\} = I$. The commonly used whitening method is eigenvalue decomposition (EVD), and the data can be whitened as:

$$\begin{aligned}\widetilde{\mathbf{x}} &= \mathbf{E}\mathbf{D}^{-1/2}\mathbf{E}^{\mathbf{T}}\mathbf{x} \\ \mathbf{D}^{-1/2} &= diag(d_1^{-1/2}, d_2^{-1/2}, \ldots, d_n^{-1/2})\end{aligned} \quad \text{,} \tag{22}$$

where $\mathbf{E}$ is the orthogonal matrix of the eigenvectors of the covariance matrix, $d_i$ is the eigenvalue.

Whitening can reduce the number of parameters to be estimated, which leads to reducing the complexity of the ICA problem. Additionally, whitening can also reduce the noise effect and prevent overfitting by reducing the dimensionality of the data during the whitening process.

The principle of FastICA is to find a vector $\mathbf{w}$ that makes $\mathbf{w}^{\mathbf{T}}\mathbf{x}$, owning the most non-Gaussianity [21]. The basic steps of univariate FastICA are as follows:

Step 1: Select an initial (random) weight vector $\mathbf{w}$;

Step 2: Let $\mathbf{w}^+ = E\{\mathbf{x}g(\mathbf{w}^{\mathbf{T}}\mathbf{x})\} - E\{g'(\mathbf{w}^{\mathbf{T}}\mathbf{x})\}\mathbf{w}$, where $g(u) = u^3$;

Step 3: Let $\mathbf{w} = \mathbf{w}^+/\|\mathbf{w}^+\|$;

Step 4: If convergent, i.e., the dot product of new and old w (almost) equals 1, end step, otherwise return to Step 2.

When estimating several independent components, the output after each iteration $(\mathbf{w}_1^{\mathbf{T}}\mathbf{x}, \mathbf{w}_2^{\mathbf{T}}\mathbf{x}, \ldots, \mathbf{w}_n^{\mathbf{T}}\mathbf{x})$ needs to be decorrelated to prevent these vectors from converging to the same maximum value. The simplest decorrelation method is the scaling method based on the Gram–Schmidt procedure, i.e., when estimating $\mathbf{w}_{\mathbf{p+1}}$, subtract the projection of the already-estimated p vectors $\mathbf{w}_1, \mathbf{w}_{\mathbf{p}}, \ldots, \mathbf{w}_{\mathbf{p}}$, and then normalize it.

Combining the above decorrelation algorithm with the univariate FastICA algorithm yields the multivariate FastICA algorithm, whose flowchart is shown in Figure 3. The independent components obtained by FastICA are the required respiratory/heartbeat signals of multiple people. Then, the respiration rate/heartbeat rate of each person can be obtained by performing FFT on each component and taking the peak value.

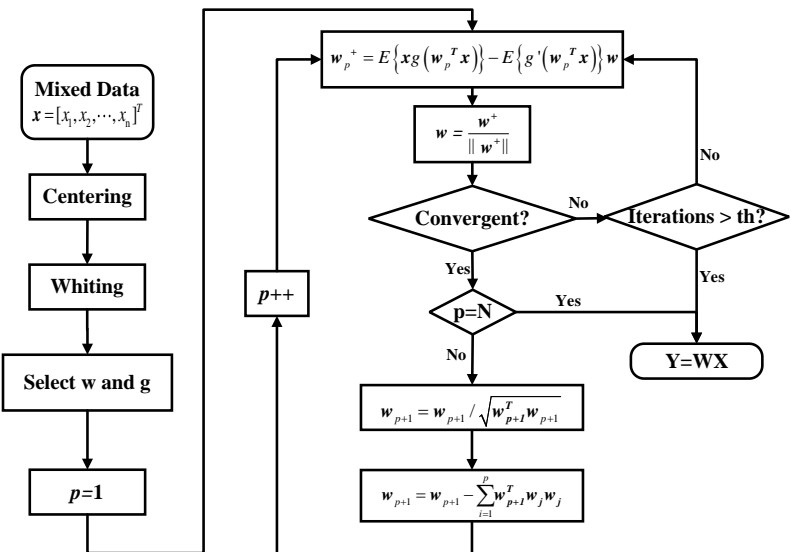

**Figure 3.** Multivariate FastICA algorithm process.

## 4. Experimental Analysis

### 4.1. Equipment Introduction

This study used the AWR1642BOOST-ODS radar device [22] and the DCA100 evaluation module (EVM) produced by Texas Instruments (TI). The device is shown in Figure 4.

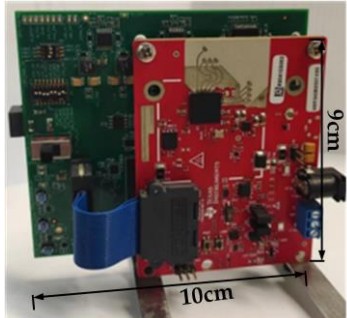

**Figure 4.** Connection diagram of radar module and data acquisition module.

The AWR1642BOOST-ODS radar parameters were set as shown in Table 1. In addition, the AWR1642BOOST-ODS was configured with two transmitting antennas and four receiving antennas. When the system operated in time-division multiplexing (TDM) MIMO mode, a non-uniform composite array composed of eight antennas was virtually generated [23]; this study only applied its four horizontal channels.

**Table 1.** Radar parameters.

| Parameter Name | Value |
| --- | --- |
| Number of Transmitting Antennas | 2 |
| Number of Receiving Antennas | 4 |
| Frequency | 77 GHz |
| Sweep Slope | 105 MHz/us |
| ADC Sampling Points | 256 |
| ADC Sampling Rate | 10 MHz |
| Bandwidth | 2.688 GHz |
| Frame Period | 40 ms |
| Number of Frames | 1500 |

In order to verify the validity and accuracy of the separation and extraction of vital signs, it is necessary to make use of other equipment that can accurately measure human breathing and heartbeat signals in the medical field. This study selected two relatively accurate contact-type devices: an abdominal belt respiratory sensor (HKH-11C) [24] and an infrared pulse sensor (HKG-07C) [25], respectively, for collecting reference signals for breathing and heartbeat.

### 4.2. Scene Setup and Data Acquisition

If there were two people nearby each other in the scenario, with a distance of 1.5 m from the radar, the scene was as shown in Figure 5. The receiving information of multiple channels of a single radar was used. Due to the size of the human body—for example, the shoulder width of an adult is about 36 cm—when two people sit side by side facing the radar at a distance of 1.5 m, the length across the radial distance of the radar is about 4.25 cm, which is smaller than the distance resolution of the radar used in this study, which was 5.58 cm. As shown in Figure 6, at this time, the two people were located in the same distance resolution unit and could not be distinguished from the spatial envelope, which means that the traditional methods based on the distance dimension or based on DOA were no longer applicable.

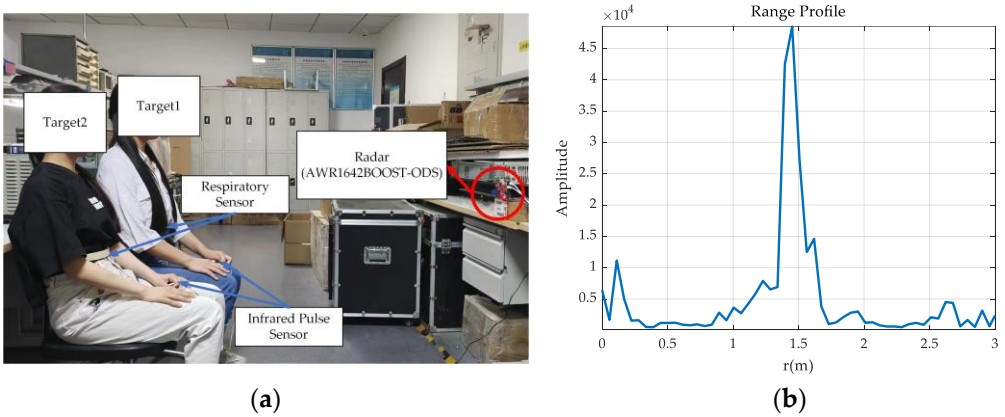

(**a**)  (**b**)

**Figure 5.** Experimental scene: (**a**) Adjacent multi-target experimental scene; (**b**) the range profile of radar echo signal.

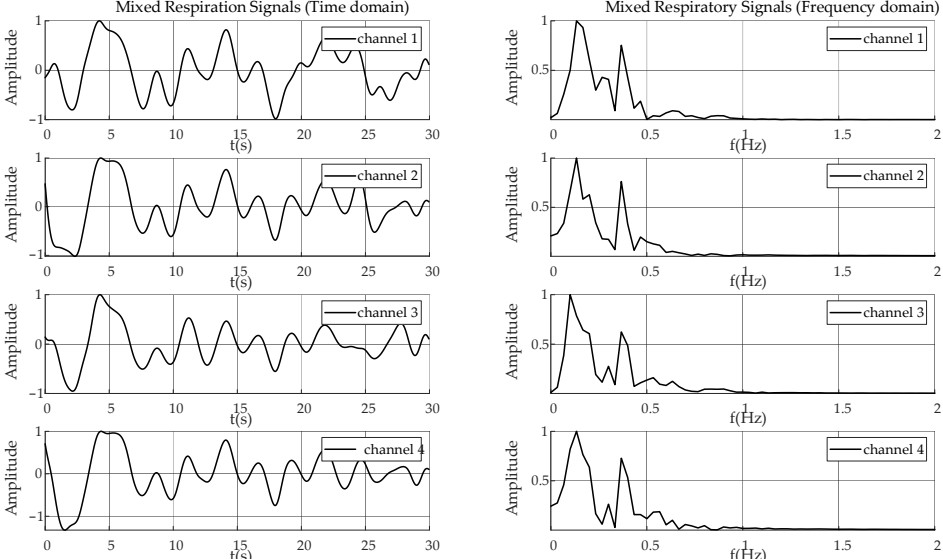

**Figure 6.** Mixed respiratory signal time domain of different channels.

### 4.3. Results and Analysis

4.3.1. Simple Scenario

For the simplest scene where two people faced the radar side by side, the method proposed in this paper was compared with the algorithms from other literature. Figure 6 shows the mixed respiratory signal of multiple people extracted from different radar channels and Figure 7 shows the separation result of different methods.

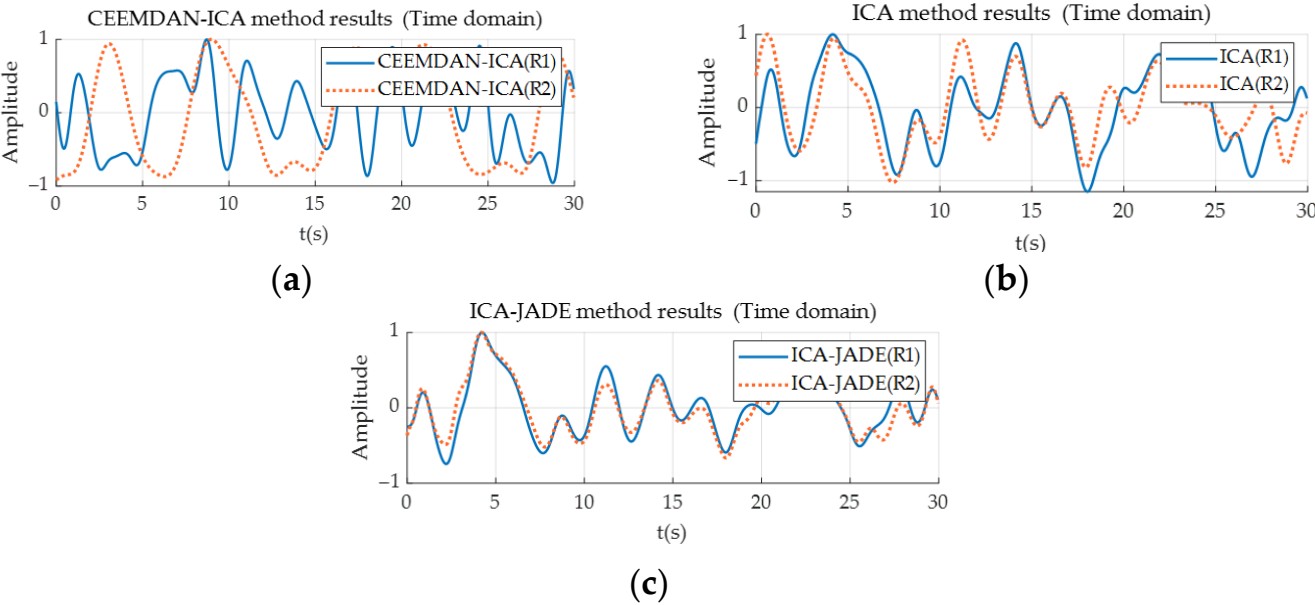

**Figure 7.** Time Domain Results of Different Methods: (**a**) CEENDAN-ICA method; (**b**) ICA method in [19]; (**c**) ICA-JADE method in [26].

It can be seen that the spatial arrangement of the radar antennas used in this study was too close, which led to the signals of different channels being similar: the time domain of the four channels' mixed respiratory signals appear to have consistent ups and downs and highly consistent periods, as well as also being concentrated in the same frequency band in the frequency domain. The ICA method proposed in [19] and the ICA-JADE method proposed in [26] failed to separate the respiratory signals of two people, as Figure 7a shows: the two-way (R1 and R2) results output by these two methods are highly consistent. The CEEMDAN-ICA method was proposed in this paper to solve this problem of the lack of effective channels. As shown in Figure 7, the proposed method successfully separated two completely different signals.

Next, we compared one of the results of the scene where two people were side by side, as shown in Figure 8a,c, with the reference signal. In Figure 8, the following can be observed from the trends of the respiratory time-domain waveforms of the two results and the corresponding reference signals: the first result, respiration CEEMDAN-ICA(T1), can fit well with target 1's reference respiratory waveform, that is, the result is valid and reliable. At the same time, the second result does not fit well with the reference respiration signal of target 2 in the period of 5 to 10 s. However, looking at the entire period, the peak values of the separated signal are equal to those of the reference signal and, at the same time, the peak values of these two are also very close in the frequency domain. The main method of general vital sign detection equipment to estimate the value of specific signs is to calculate the number of peaks within a period of time (peak counting method) or to take the frequency of the peak of the spectrum (spectrum method). As shown by the experimental results displayed in Figure 8, the frequencies of the respiratory signals of the two targets separated by the CEEMDAN-ICA method are 0.55 Hz and 0.4 Hz and the frequencies of the heartbeat signals are 1.13 Hz and 1.03 Hz, respectively. Comparing the estimation results with the reference signals, the absolute value errors of the respiratory

rates of the two targets are 1.02 breaths/minute and 1.38 breaths/minute, and the absolute value errors of the heart rates are 3.6 beats/minute and 0.6 beats/minute. Table 2 shows the respiration and heartbeat estimation results and errors of five experiments of the proposed CEMDAN-ICA method in the simple scene shown in Figure 8a,c. At this time, it can be calculated that the average breathing error is 1.24 breaths/minute and the average heartbeat error is 2.63 beats/minute.

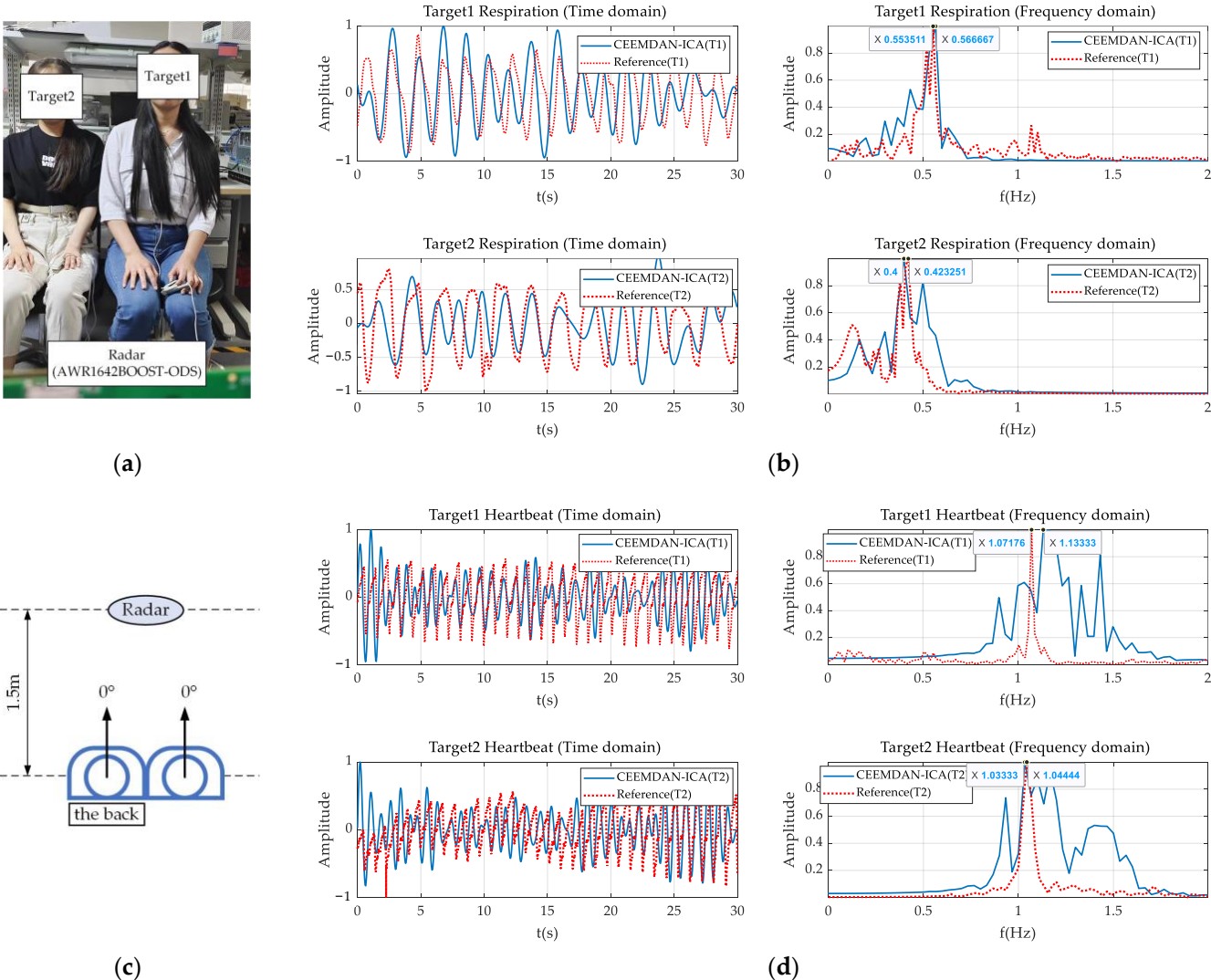

**Figure 8.** Signal separation results of CEEMDAN-ICA method for the 2-person scene: (**a**) (0°, 0°) experimental scene photo; (**b**) The comparison between the respiratory signals of 2 human targets (results of CEEMADN-ICA method) and reference signals in the time domain and frequency domain; (**c**) (0°, 0°) experimental schematic diagram; (**d**) The comparison between the heartbeat signals of 2 human targets (results of CEEMADN-ICA method) and reference signals in the time domain and frequency domain.

**Table 2.** Separation results and errors of CEEMDAN-ICA method in simple scenario.

| Num | Target ID | RR * (bpm) | RR * (Ref) (bpm) | RR * Errors (bpm) | HR * (bpm) | HR * (Ref) (bpm) | HR * Errors (bpm) |
|---|---|---|---|---|---|---|---|
| 1 | (1) | 34.20 | 33.18 | 1.02 | 67.80 | 64.20 | 3.6 |
|   | (2) | 24.00 | 25.38 | 1.38 | 61.80 | 62.40 | 0.6 |
| 2 | (1) | 19.80 | 18.18 | 1.62 | 63.60 | 62.64 | 0.96 |
|   | (2) | 22.20 | 22.80 | 0.6 | 66.00 | 63.30 | 2.7 |
| 3 | (1) | 13.20 | 13.80 | 0.6 | 69.60 | 66.60 | 3 |
|   | (2) | 25.80 | 27.60 | 1.8 | 60.00 | 64.20 | 4.2 |
| 4 | (1) | 12.00 | 13.20 | 1.2 | 63.60 | 68.40 | 4.8 |
|   | (2) | 25.80 | 24.00 | 1.8 | 61.80 | 66.60 | 4.8 |
| 5 | (1) | 16.20 | 15.00 | 1.2 | 67.80 | 66.36 | 1.44 |
|   | (2) | 21.60 | 22.80 | 1.2 | 64.20 | 64.02 | 0.18 |

* RR is Respiration Rate, HR is Heartbeat Rate.

### 4.3.2. Scenarios of Subjects with Different Orientations

This paper set up multiple experimental scenarios to verify the effectiveness of the algorithm. In addition to the simple experimental scene (that is, two people sitting side by side and facing the radar), as shown in Figure 8a,c, we added three other scenes: one person facing the radar and the other at a 45° angle side facing the radar, which is shown in Figure 9a,c; one person facing the radar and the other facing the radar at a 90° angle, which is shown in Figure 10a,c; two people facing the radar at −90° and 90°, respectively, that is, two people back to back and facing the radar sideways, which is shown in Figure 11a,c.

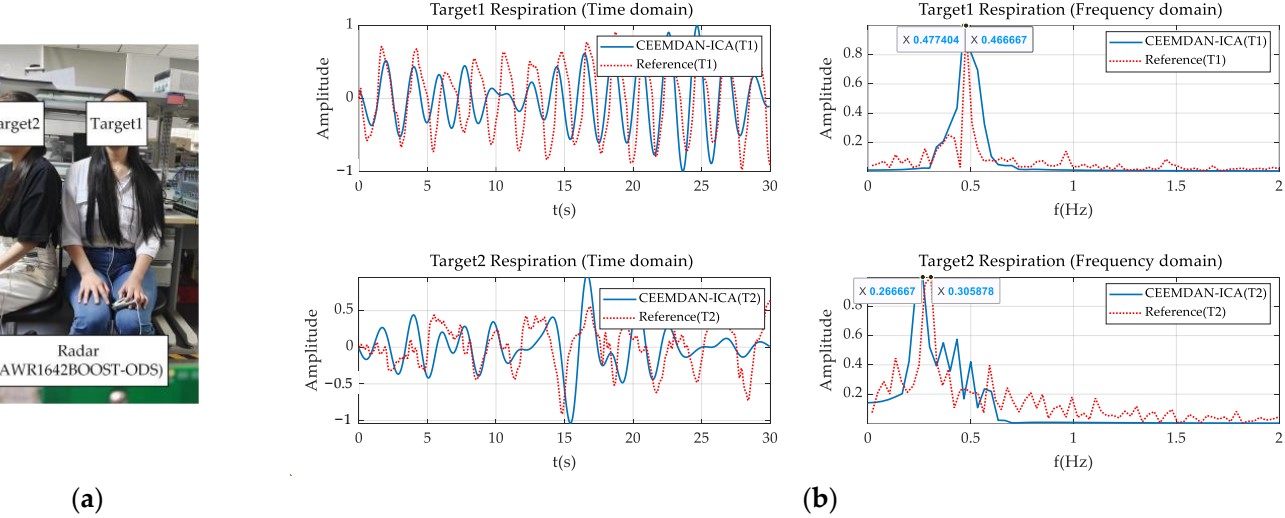

(**a**)  (**b**)

**Figure 9.** *Cont.*

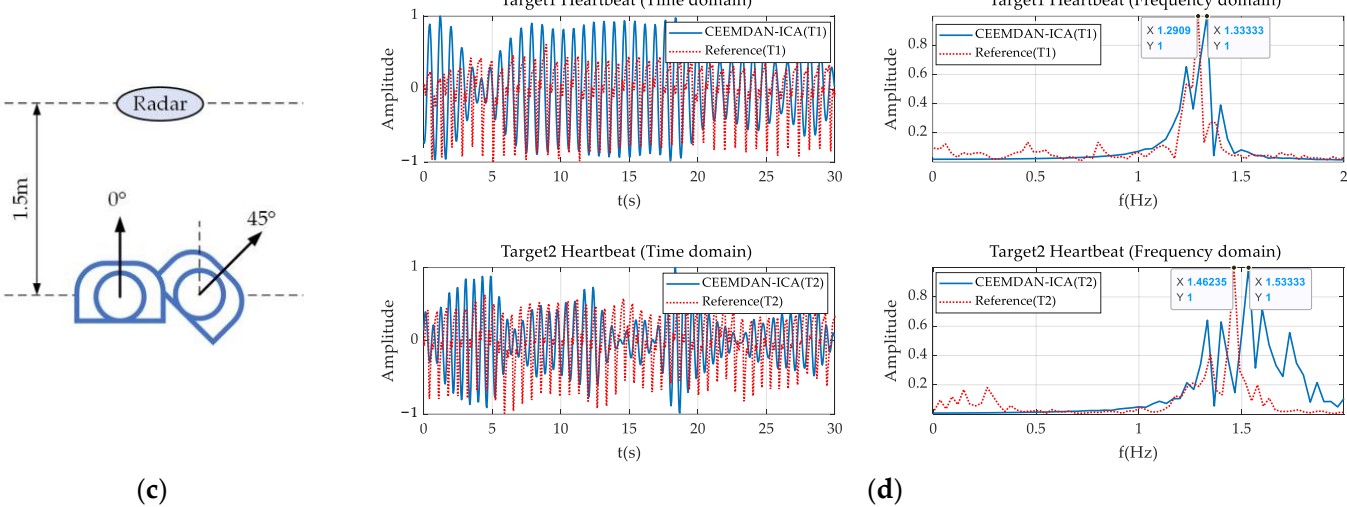

**(c)**

**(d)**

**Figure 9.** Signal separation results of CEEMDAN-ICA method for 2-person scene: (**a**) (0°, 45°) experimental scene photo; (**b**) The comparison between the respiratory signals of 2 human targets (results of CEEMADN-ICA method) and reference signals in time domain and frequency domains; (**c**) (0°, 45°) experimental schematic diagram; (**d**) The comparison between the heartbeat signals of 2 human targets (results of CEEMADN-ICA method) and reference signals in time domain and frequency domains.

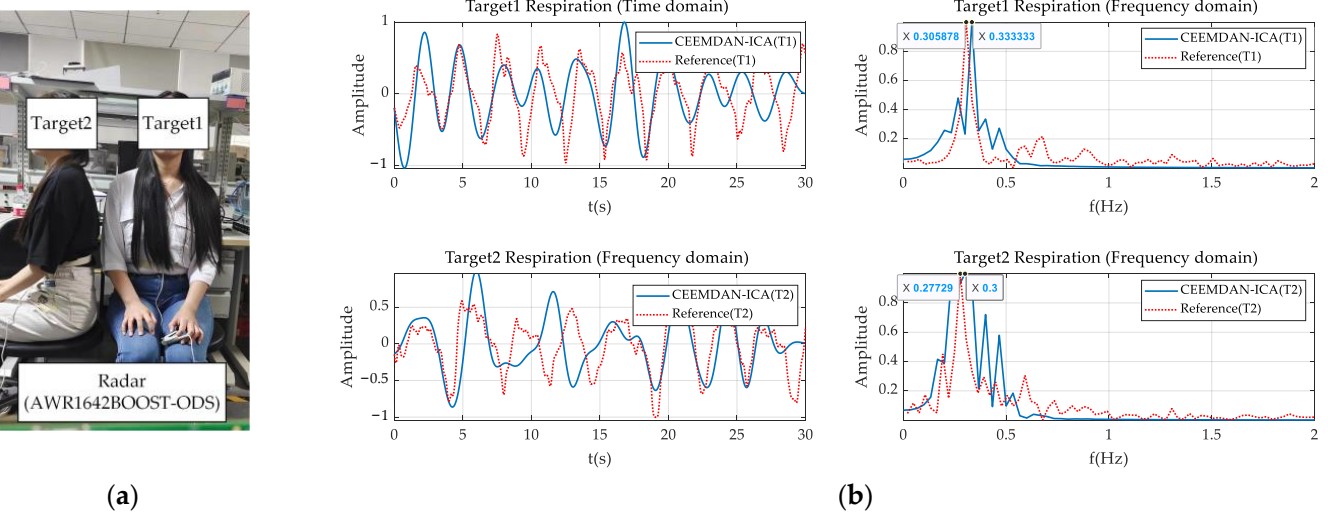

**(a)**

**(b)**

**Figure 10.** *Cont.*

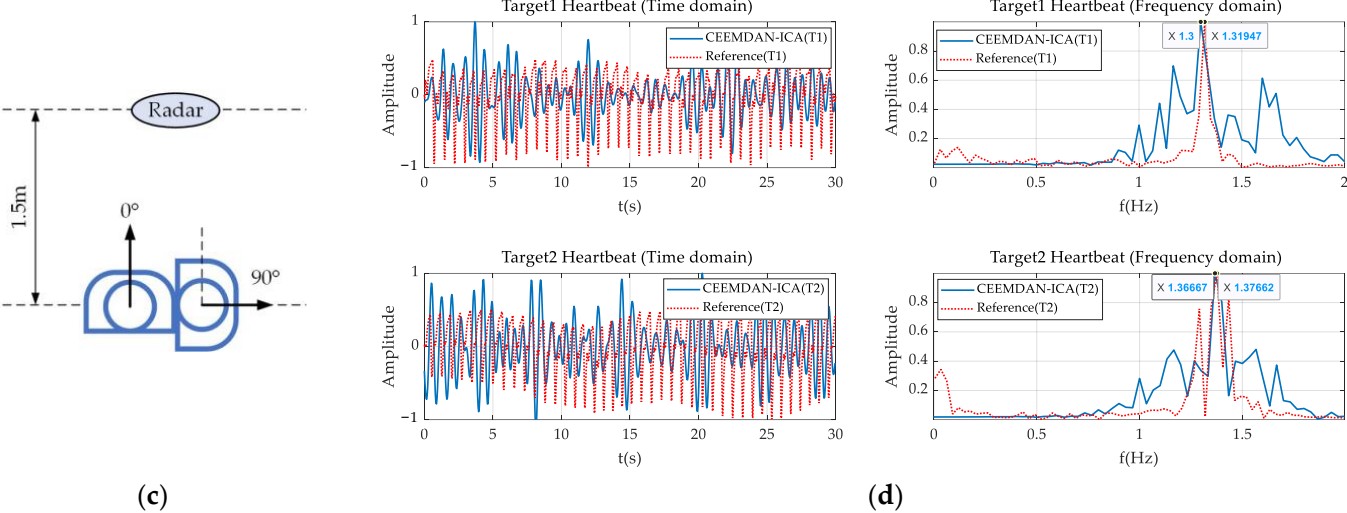

**Figure 10.** Signal separation results of CEEMDAN-ICA method for 2-person scene: (**a**) (0°, 90°) experimental scene photo; (**b**) The comparison between the respiratory signals of 2 human targets (results of CEEMADN-ICA method) and reference signals in the time domain and frequency domain; (**c**) (0°, 90°) experimental schematic diagram; (**d**) The comparison between the heartbeat signals of 2 human targets (results of CEEMADN-ICA method) and reference signals in the time domain and frequency domain.

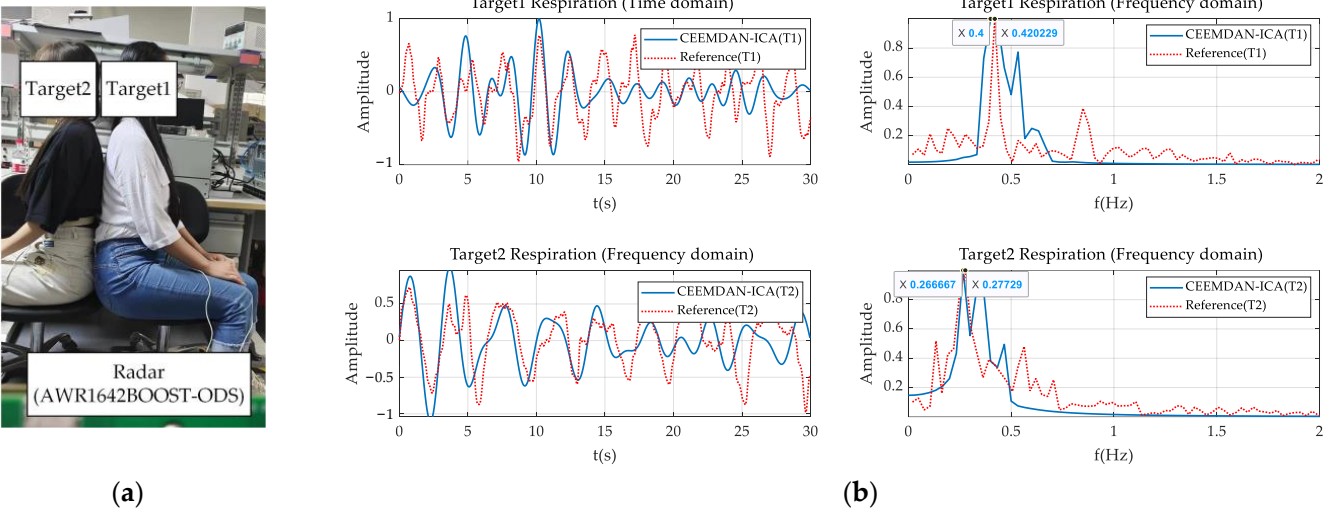

**Figure 11.** *Cont.*

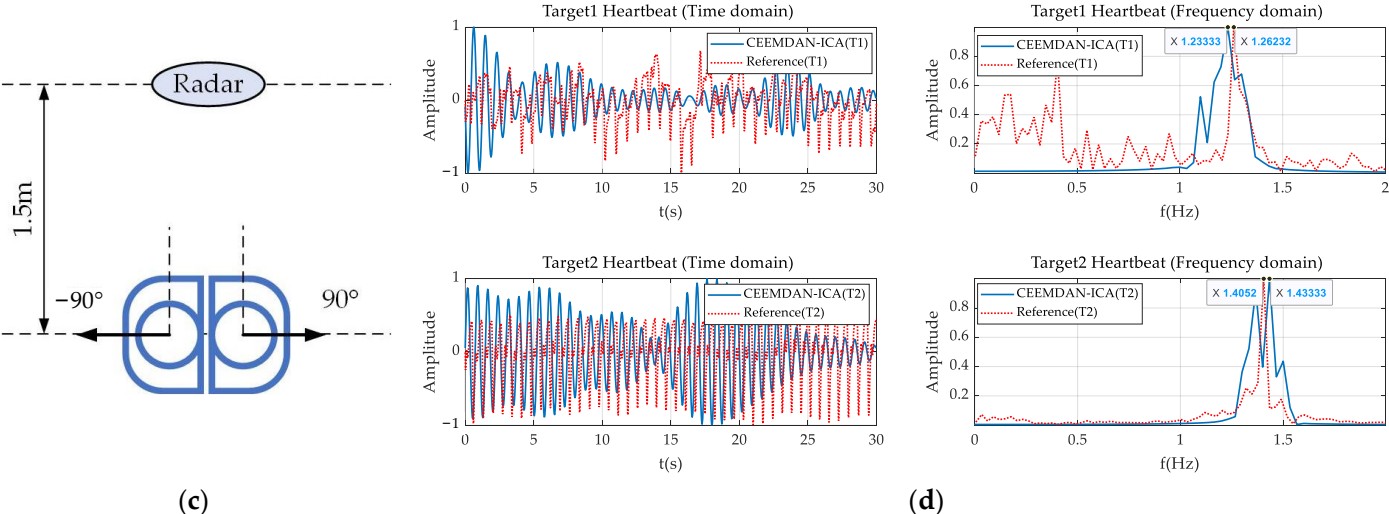

**Figure 11.** Signal separation results of CEEMDAN-ICA method for the 2-person scene: (**a**) (−90°, 90°) experimental scene photo; (**b**) The comparison between the respiratory signals of 2 human targets (results of CEEMADN-ICA method) and reference signals in the time domain and frequency domain; (**c**) (−90°, 90°) experimental schematic diagram; (**d**) The comparison between the heartbeat signals of 2 human targets (results of CEEMADN-ICA method) and reference signals in the time domain and frequency domain.

Comparing the three scenarios in Figures 9–11, it can be seen that the time-domain waveforms of breathing or heartbeat obtained in all scenarios can fit well with the reference waveform, which proves the effectiveness and reliability of the algorithm. It is worth mentioning that, generally, the human body facing the radar is the most ideal scene for breathing and heartbeat extraction. In the scene in Figure 11, two people are facing the radar back-to-back. At this time, the spectral width of breathing obtained by comparing different scenes does not change much. However, the spectral width of the heartbeat is narrowed. This may be because, when the human body is turned sideways, the breathing amplitude detected by the radar becomes smaller, that is, the heartbeat amplitude will be more obvious. At the same time, in the scene where the left side of the body is close to the target of the radar—here it is target 2, since the heart is located on the left side of the body—the heartbeat information will be more obvious, and it can be seen that the spectrum of T2 is indeed narrower. Above all, we can see that the method proposed in this paper can effectively separate the respiratory and heartbeat signals of two people in different scenarios.

In order to analyze the reliability of the results, the concept of mean accuracy is introduced:

$$\text{Accuracy} = \frac{1}{N}\sum_{i=1}^{N}\left(1 - \frac{|e_i - g_i|}{e_i}\right) \times 100\%, \tag{23}$$

where $e_i$ is the estimated rate from the proposed method, $g_i$ is the ground truth frequency, which is obtained by the respiratory belt or infrared pulse sensor, and N is the number of experiments.

Table 3 shows the mean respiration rate (RR) accuracy and mean heartbeat rate (HR) accuracy obtained in the experiments of the two-person scene. It can be seen that, in the four experimental scenes, the method proposed in this paper successfully separated the respiration and heartbeat signals of two people. The accuracy of the extracted vital sign signals is the highest in the scene where the two people face the radar side by side. At this time, the mean RR accuracies of target 1 and target 2 are 95.55% and 94.99%, and the mean HR accuracies are 98.18% and 97.01%. However, when the two targets face the radar back to back from the (−90°, 90°) orientation, the accuracy drops. The RR and HR accuracy of

Target1 drops to 88.94% and 93.82%, respectively. In the scene where the human body is not directly facing the radar, the accuracy will decrease. This is mainly due to the change in the detectable amplitude of the human chest displacement in different orientations. By averaging the results of all experimental scenarios, the overall mean accuracy for the breathing and heartbeat of the method proposed in this paper is 92.24%.

**Table 3.** Mean accuracy of RR and HR under different scenes.

| Seq. | Scenes | RR Accuracy | | HR Accuracy | |
|---|---|---|---|---|---|
| | | Target1 | Target2 | Target1 | Target2 |
| (a) | $(0°, 0°)$ | 95.55% | 94.99% | 98.18% | 97.01% |
| (b) | $(0°, 45°)$ | 94.71% | 91.67% | 96.18% | 97.00% |
| (c) | $(0°, 90°)$ | 93.29% | 87.26% | 97.11% | 97.54% |
| (d) | $(-90°, 90°)$ | 88.94% | 91.55% | 93.82% | 97.31% |
| Mean | | 92.24% | | 96.77% | |

## 5. Conclusions

This paper proposed a radar multi-target vital signs detection method based on CEEMDAN-ICA, which overcomes the limitation of detecting vital signs of each human target when multiple human targets exist within the same range and angular resolution cell. The proposed method fully utilizes the motion information of the human body, converts the human chest motion features into virtual multi-angle observation results using the CEEMDAN algorithm, and then applies the FastICA algorithm to obtain the breathing rate/heart rate of each individual. The processed experimental data results show that only the proposed method can obtain multiple targets' vital signs detection results within the same resolution cell. Four different adjacent multi-target scenarios were tested and the experiments proved that the proposed method achieves a 92.24% accuracy for respiratory rate and a 96.77% accuracy for heart rate signal detection.

**Author Contributions:** X.D. and Y.F. had the research idea, were involved in the research design and contributed to revising the manuscript. Y.F. and C.C. performed the simulation and wrote this manuscript. Y.F., C.C. and J.L. contributed to revising and improving the research. All authors have read and agreed to the published version of the manuscript.

**Funding:** This research was funded in part by the Distinguished Young Scholars of Chongqing (Grant No. cstc2020jcyj-jqX0008) and the Postdoctoral Science Foundation of Chongqing in China (2022NSCQ-BHX5720).

**Data Availability Statement:** Not applicable.

**Conflicts of Interest:** The authors declare no conflict of interest.

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
