# Peer review of "CEEMDAN-ICA-Based Radar Monitoring of Adjacent Multi-Target Vital Signs"

_electronics, doi:10.3390/electronics12122732_

Round 1

Reviewer 1 Report

This paper introduces a CEEMDAN-ICA method to detect respiratory rate and heart rate in multiple targets close to each other. I have several concerns over the experimental design and manuscript preparation, due to the nature of these concerns, I do not recommend publication in the current format.

Major concern: In this experiment, the authors only tested one position where two targets were sitting side by side and facing the radar. Will the accuracy of this model be changed when the targets change their positions? e.g., two targets are sitting back-to-back; the radar is not facing directly to the targets. Any difference if two targets are not sitting together? what happened if the targets are moving? Could you measure more than two targets?

Other concerns:

1.     The authors need to justify why they want to measure multiple targets close to each other. What is the significance?

2.     Describe the full name when first using an abbreviation. Such as, the CEEMDAN-ICA, LFMCW, LMS

3.     Introduce more about filter design methods (butterworth, Chebyshev type I & II, and elliptic).

4.     Add appropriate citations in the introduction/modeling section. The citation format need to be more professional: instead of using “MIT proposed a vital-radio”, the authors should use “Adib et al. proposed” or “Miller groups proposed”.

5.     Figure 5 & 6 is labeled as Figure 2.

Reviewer 2 Report

Comments:

1. The article has a high similarity with other works.

2. There is a lack of comments on the sample of experiments.

3. The results presented are too few for the quality of your research.

4. It does not present a comparison with other systems developed and published.

5. You quote an accuracy of 94% y 96% with respect to what you can buy or the statement how did you make it?

Round 2

Reviewer 1 Report

The authors have addressed most of my concerns.  The authors claimed that existing methods can be used to detect the vital signals of a couple or a new mother sleeping with her infant in the same bed, but they still miss the “why”, what is the significance of multi-adjacent-target signal detection? This claim brought a new question about the accuracy when measuring two targets with large size differences (Mother & Infant).

Other minor revisions:

1.       In Figure 4, add a scale bar.

2.       Uniform the capitalization in Figure 5 (respiratory sensor, infrared pulse sensor, (b) the range…)

3.       Increase the font size of Figure 6.

4.       Add coordinate axis to Figure 8c,  9c, 10c, 11c.

Reviewer 2 Report

Comments:

1. I update the abstract of the article, giving a good focus with this.

2. Added some words and paragraphs in the introduction section.

3. Section 2 changed slightly with the additions.

4. Section 3 didn't change much, just added some comments.

5. In section 4.2 I only added one more figure, it could only justify your contribution.

6. The results and analysis section is very well fed.

7. Table 2 should be added because it is complementary to your evidence.

8. Missing answer: You quote an accuracy of 94% and 96% with respect to what you can buy or the statement how did you make it?
